# A population-based retrospective study comparing cancer mortality between Moluccan migrants and the general Dutch population: equal risk 65 years after immigration?

Junus M. van der Wal, Adee Bodewes, Charles Agyemang, Anton Kunst

**To cite:** van der Wal JM, Bodewes A, Agyemang C, et al. A population-based retrospective study comparing cancer mortality between Moluccan migrants and the general Dutch population: equal risk 65 years after immigration? *BMJ Open* 2019;**9**:e029288. doi:10.1136/bmjopen-2019-029288

Department of Public Health, Amsterdam UMC (location AMC), University of Amsterdam, Amsterdam, The Netherlands

**Correspondence to**
Junus M. van der Wal;
j.m.vanderwal@amc.uva.nl

## ABSTRACT

**Objective** To test the hypothesis that cancer mortality rates among the Moluccan–Dutch, the oldest non-Western migrant group to arrive in the Netherlands after the Second World War, are similar to those in the general Dutch population.

**Design** Population-based retrospective study.

**Setting** Data from the national cause of death registry in the Netherlands and municipal registries.

**Participants** Using historic records containing family names of all Moluccan–Dutch who arrived in the Netherlands in 1951, we identified 81 591 Moluccan–Dutch persons in the national cause of death registry of the Netherlands. The reference group consisted of 15 866 538 persons of the general Dutch population.

**Outcome measures** Mortality data were linked to demographic data from municipal registries. We calculated all-cancer and cancer-specific mortality and measured differences between the two groups using Poisson regression, adjusting for sex, age and area socioeconomic status. We conducted a sub-analysis for the first-generation and second-generation Moluccan–Dutch.

**Results** There was no difference in all-cancer mortality between Moluccan–Dutch and the general Dutch population. Mortality was higher among Moluccan–Dutch for liver, cervix and corpus uteri cancers, but lower for stomach, oesophagus, kidney and nervous system cancers. For most cancers, mortality risk as compared with the general Dutch population varied between different generations of Moluccan–Dutch.

**Conclusions** Several decades after migration, the Moluccan–Dutch show similar all-cancer mortality, but different cancer-specific mortality rates, when compared with the general Dutch population.

## INTRODUCTION

Ethnic disparities in the prevalence and mortality burden of different cancer types are known to exist in Europe.[1] For instance, migrants from low-income and middle-income countries (LMIC) tend to show lower all-cancer mortality and morbidity. Moreover, cancer burden among LMIC-migrants

## Strengths and limitations of this study

► This is the first study to investigate cancer mortality among the Moluccan–Dutch, the oldest non-western migrant group to enter the Netherlands after the Second World War.

► We employed a novel approach to select persons from Moluccan descent from the Dutch national cause of death registry, by selecting persons based on the Moluccan family names registered in historical immigration documents, which were made available to us for research purposes.

► A limitation of this novel selection method is the fact the we were unable to identify second-generation Moluccan–Dutch persons with a native Dutch father and consequently Dutch family name.

► Another limitation is the low number of deaths for most cancer types in the Moluccan–Dutch group, limiting statistical power.

is thought to be influenced by differences in cancer epidemiology between high-income countries (HICs) and LMIC and exposure to concordant risk factors.[2] When compared with their host population, cancers related to infection, such as liver, stomach and cervical cancer, are more common in migrants from LMICs; cancers considered to be related to Western lifestyle such as colorectal, breast and oesophageal cancer are less common in these groups.[1]

Increasing evidence suggests that such epidemiological differences become less pronounced as LMIC-migrants reside in their destination country for a longer period, as local (Western) cancer-related risk factors will eventually predominate over risk factors associated with their country of origin.[2] Moreover, second-generation migrants who were born in HICs lack early life risk factor exposures experienced by the first generation, such as possible higher infection

pressure.[3] Accordingly, some studies on LMIC-migrants who have resided in their host country for more than one generation have suggested a shift in cancer epidemiology towards rates of the native population of their destination country.[4–6] In the Netherlands, such a pattern has been shown most consistent in Surinamese migrants for mortality caused by most major cancers, but is also suggested for migrants from Turkey, Morocco and the Netherland Antilles.[4 6] Given the limited number of studies on this topic, more research is necessary to further characterise the cancer risk profile in these groups and address the question to what extent ethnic disparities in cancer mortality persist or start to converge to those of the host population.

For this reason, we set out to assess the difference in cancer mortality between the general Dutch population and the Moluccan–Dutch, the oldest LMIC migrant group to settle in the Netherlands after the Second World War. The Moluccan–Dutch are a migrant group consisting of the former soldiers from the Royal Netherlands East Indies Army (KNIL), the military force of the Netherlands in their former colony. These soldiers, and their families, were transported to the Netherlands in spring 1951 after decolonisation of Indonesia. Even though their stay was supposed to be temporary, the majority settled in the Netherlands after their struggle to found their own republic on the Moluccan Islands failed. No considerable chain or remigration occurred, resulting in a closed minority group, the vast majority of which resided in segregated Moluccan districts throughout the Netherlands.[7]

Earlier studies found differences in cancer standard incidence ratios and mortality between migrants from Indonesia and the general Dutch population.[8–10] A separate analysis of the Moluccan–Dutch is desirable, since their particular migration history resulted in a homogeneous group within the Indonesian community with their own cultural practices, lifestyle and migration history, all of which can have a profound impact on cancer risk.

The aim of this study was to assess the difference in all cancer and cancer-specific mortality rates between the Moluccan–Dutch and the general Dutch population, with a subanalysis for the first and second generations. Since the Moluccan–Dutch have resided in the Netherlands for over 65 years, we expect local environmental factors in the Netherlands to reflect on mortality rates, for instance in cancers that are known to be influenced by factors such as lifestyle or exposure to infectious agents. Therefore, we hypothesise cancer mortality rates to be similar to those of the host population, especially for the second generation.[4 6]

## METHODS
### Data sources and handling
For this study, we used mortality data as recorded in the nationwide cause of death registry of Statistics Netherlands (CBS) between the years 2000 and 2013. Deaths in the Netherlands are documented by a physician on a death certificate form and cause of death is classified in accordance with the 10th Revision of the International Classification of Diseases (ICD-10), after which it is recorded in the cause of death registry. Demographic data were derived from municipal registries and included date of birth, sex and place of residence. Both registries include all legal residents in the Netherlands, and are linked through personal identification numbers.

Moluccan–Dutch persons were identified in the registries based on surname. Using passenger lists of the various boats that transported Moluccans to the Netherlands in 1951, the Museum Maluku Netherlands created a list containing approximately 1700 surnames of the arriving Moluccan families. This list was used to identify and link persons of Moluccan descent in these registries. After identification and linking, data were anonymised. The reference group, referred to as 'general Dutch population' consisted of 15 866 538 persons with Dutch nationality.

We used birth year to determine Moluccan generations. We considered people born up to 1951, the year of migration, to be of the first generation and people born after 1951 but before 1970 of the second generation. We chose 1970 as cut-off point and excluded persons born hereafter, since Moluccan–Dutch born after this year will increasingly be offspring of the second generation (ie, the third generation) and because the number of death in this group was low due to relatively young age. Furthermore, we created two groups within the first generation, based on whether the person was an adult (born before 1934) or a minor (born between 1934 and 1951) in the year that the Moluccan migrants were transported to the Netherlands. We made this stratification, since Moluccan–Dutch who migrated to the Netherlands as minor are expected to show distinctly different cancer risk exposures than both the first-generation adults, who have similar early life exposures from Indonesia but are less integrated in Dutch society, or the second generation, who lack early life exposures from Indonesia altogether.

In one analysis, we only included Moluccan–Dutch of the first and second generations, and we compared them to their Dutch counterparts born in the same periods. This reference group included 9 064 169 persons with Dutch nationality born up to 1970.

Area-level social economic status (SES) was used as proxy for individual SES. Area-level SES was based on decile scores calculated by the Netherlands Institute for Social Research (SCP), reflecting SES of inhabitants of specific postal codes. A higher decile score indicates a higher area SES.

### Data analysis
All-cancer mortality encompassed codes C00-D48 in the ICD-10 and Related Health Problems. Furthermore, we chose to investigate the most common types of cancer in both HICs and LMICs, as identified by the global cancer research project GLOBOCAN 2012.[11] In breast cancer,

we only included female patients due to low number of deaths among males for this cancer type.

We present both absolute numbers of deaths, as well as age-adjusted death rates, using the direct method of standardisation. In accordance with CBS guidelines, number of deaths lower than 10 are not shown. Poisson regression analysis was used to investigate the size of difference in mortality attributable to the different types of cancer between Moluccan–Dutch and the general Dutch population, adjusted for sex, age-at-baseline (age in the year 2000, beginning of mortality data collection) and SES. Furthermore, we performed separate Poisson regression analyses for the first and second-generation Moluccan–Dutch and for each sex. Based on these regression analyses, we present relative risks (RR) with 95% CI. Analyses were performed using IBM SPSS Statistics V.20.

### Patient and public involvement

We used data from existing databases. No patients were involved in the development of the research question, design of the study and sourcing of the data. Results of this study will be disseminated among the Moluccan–Dutch communities in the Netherlands via social media outlets.

## RESULTS

Demographic characteristics of our study population are presented in table 1. A total of 81 591 Moluccan–Dutch persons were included in this study. Within the Moluccan–Dutch group, 15.7% was of the first generation (5.7% migrated as adult and 10.0% as minor), 27.9% was of the second generation and 56.4% was of the second or third generation. The Moluccan–Dutch in our study population showed higher rates in the lower SES groups, when compared with the general Dutch population.

Table 2 gives the overview of the investigated cancer types and the corresponding total number of deaths and age-adjusted death rates, as well as all-cancer mortality. In both the Moluccan–Dutch and the general Dutch population groups, highest age-adjusted death rates were found for lung cancer, followed by colorectal and breast cancer, respectively.

RR of cancer mortality for each investigated cancer type is presented in table 3. The Moluccan–Dutch showed similar all-cancer mortality risk when compared with the general Dutch population (RR=0.98, CI: 0.93 to 1.03). However, Moluccan–Dutch men showed lower all-cancer mortality than men in the general Dutch population (RR=0.92, CI: 0.85 to 0.99). Furthermore, the Moluccan–Dutch showed

| Table 1 | Demographic characteristics of the Moluccan–Dutch and the Dutch general population | | | |
|---|---|---|---|---|
| | **Moluccan–Dutch** | | **Dutch general population** | |
| | **N** | **%** | **N** | **%** |
| Sex | | | | |
| Male | 40 062 | 49.1 | 7 917 402 | 49.9 |
| Female | 41 529 | 50.9 | 7 949 136 | 50.1 |
| Generation (birth year) | | | | |
| 1 adult (<1934) | 4642 | 5.7 | 2 480 434 | 15.6 |
| 1 minor (1935–1951) | 8185 | 10.0 | 2 639 408 | 16.6 |
| 2 (1951–1969) | 22 754 | 27.9 | 3 944 327 | 24.9 |
| ≥2 (1970>) | 46 010 | 56.4 | 6 802 369 | 42.9 |
| Decile score | | | | |
| 1 (lowest SES) | 7017 | 8.6 | 729 861 | 4.6 |
| 2 | 7669 | 9.4 | 951 992 | 6.0 |
| 3 | 10 363 | 12.7 | 1 729 453 | 10.9 |
| 4 | 11 341 | 13.9 | 1 745 319 | 11.0 |
| 5 | 11 096 | 13.6 | 2 491 047 | 15.7 |
| 6 | 6935 | 8.5 | 1 919 851 | 12.1 |
| 7 | 6283 | 7.7 | 1 856 385 | 11.7 |
| 8 | 4732 | 5.8 | 1 174 124 | 7.4 |
| 9 | 7262 | 8.9 | 1 586 654 | 10.0 |
| 10 (highest SES) | 8648 | 10.6 | 1 650 120 | 10.4 |
| Missing | 245 | 0.3 | 31 732 | 0.2 |
| Total | 81 591 | 100 | 15 866 538 | 100 |

SES, socioeconomic status.

**Table 2** Absolute numbers of death and age-standardised death rates per cancer type

| Type of cancer | ICD-10 code | Absolute numbers of death | | Age-standardised death rates per 100 000 | |
|---|---|---|---|---|---|
| | | Dutch general population | Moluccan–Dutch | Dutch general population | Moluccan–Dutch |
| Stomach | C16 | 18 443 | 33 | 4.98 | 3.51 |
| Liver cell | C22 | 7227 | 53 | 2.03 | 6.37 |
| Cervix uteri | C53 | 2488 | 15 | 0.77 | 1.51 |
| Colorectal | C18-21 | 59 548 | 138 | 16.11 | 15.51 |
| Breast | C50 | 41 324 | 134 | 12.59 | 13.20 |
| Prostate | C61 | 30 975 | 79 | 7.60 | 10.93 |
| Bladder | C67 | 14 875 | 27 | 3.86 | 2.91 |
| Lung and lower airway | C33-34 | 119 520 | 336 | 34.67 | 34.89 |
| Oesophagus | C15 | 18 683 | 25 | 5.51 | 2.55 |
| Corpus uteri | C54 | 5189 | 22 | 1.38 | 2.15 |
| Ovaries | C56 | 12 239 | 40 | 3.52 | 3.76 |
| Pancreas | C25 | 27 351 | 62 | 7.64 | 6.95 |
| Kidney | C64 | 11 491 | 20 | 3.24 | 1.98 |
| Leukaemia | C91-95 | 14 349 | 48 | 3.71 | 5.04 |
| Non-Hodgkin's lymphoma | C82-85, C96 | 13 620 | 48 | 3.70 | 5.13 |
| Nervous system | C70-72 | 10 893 | 19 | 3.36 | 1.07 |

ICD-10, 10th revision of the International Statistical Classification of Diseases and Related Health Problems.

lower RR for mortality caused by cancer of the stomach (RR=0.71, CI: 0.50 to 1.00), oesophagus (RR=0.46, CI: 0.31 to 0.68), kidneys (RR=0.61, CI: 0.39 to 0.96) and nervous system (RR=0.32, CI: 0.18 to 0.58), when compared with the general Dutch population. In contrast, they showed higher RR for mortality caused by cancer of the liver (RR=2.60, CI: 1.98 to 3.42), cervix (RR=1.89, CI: 1.13 to 3.14) and corpus uteri (RR=1.72, CI: 1.12 to 2.64). Additionally, only Moluccan–Dutch men showed lower mortality risk for cancer of the stomach (RR=0.57, CI: 0.35 to 0.91) and pancreas (RR=0.63, CI: 0.42 to 0.95) and higher mortality risk for non-Hodgkin's lymphoma (NHL) (RR=1.60, CI: 1.13 to 2.27), when compared with men in the general Dutch population.

A regression analysis per generation within the Moluccan–Dutch group is presented in table 4. Among the first-generation adults, stomach cancer mortality was lower (RR=0.48, CI: 0.26 to 0.89), but NHL mortality was higher (RR=1.68, CI: 1.13 to 2.51), when compared with the general Dutch population. Furthermore, oesophageal cancer mortality was lower in the first-generation adults (RR=0.37, CI: 0.17 to 0.82) and minors (RR=0.39, CI: 0.21 to 0.76), whereas liver cancer mortality was higher among the first-generation adults (RR=3.96, CI: 2.74 to 5.70) and minors (RR=1.79, CI: 1.06 to 3.03), when compared with their counterparts in the general Dutch population. Significantly elevated among the second generation, but not among first generation, were cancer of the cervix (RR=2.39, CI: 1.28 to 4.47), breast (RR=1.29, CI: 1.01 to 1.65), corpus uteri (RR=3.14, CI:

1.56 to 6.33) and ovaries (RR=1.76, CI: 1.09 to 2.83) and leukaemia (RR=1.95, CI: 1.17 to 3.25). Cancer of the brain and nervous system carried lower mortality among both first-generation minors (RR=0.39, CI: 0.16 to 0.93) and the second-generation Moluccans (RR=0.25, CI: 0.09 to 0.66), when compared with the general Dutch population.

Two cancer types show inverse outcomes when comparing mortality risk of two consecutive generations of Moluccan–Dutch with the general Dutch population. Prostate cancer mortality was higher among first-generation adults in the Moluccan–Dutch group when compared with their counterparts in the general Dutch population (RR=1.61, CI: 1.28 to 2.03), but lower among first-generation minors (RR=0.45, CI: 0.22 to 0.90). Additionally, lower risk of dying of lung cancer was observed among first-generation minors when compared with the general Dutch population (RR=0.70, CI: 0.58 to 0.85), but the second-generation Moluccan–Dutch showed a higher risk (RR=1.42, CI: 1.20 to 1.69).

## DISCUSSION
### Key findings
Our objective was to compare cancer mortality between the Moluccan–Dutch and the general Dutch population. We found similar all-cancer mortality, but several differences in cancer-specific mortality across both the first-generation and second-generation Moluccan–Dutch.

van der Wal JM, *et al. BMJ Open* 2019;**9**:e029288. doi:10.1136/bmjopen-2019-029288

**Table 3** Poisson regression for cancer-specific mortality and all-cancer mortality among Moluccan–Dutch, adjusted for age-at-baseline and socioeconomic status

| Cancer type | N | Both sexes RR (95% CI) | N | Males RR (95% CI) | N | Females RR (95% CI) |
|---|---|---|---|---|---|---|
| Stomach | 33 | 0.71 (0.50 to 1.00) | 17 | 0.57 (0.35 to 0.91) | 16 | 0.96 (0.59 to 1.57) |
| Liver | 53 | 2.60 (1.98 to 3.42) | 33 | 2.58 (1.83 to 3.64) | 20 | 2.65 (1.69 to 4.17) |
| Cervix | 15 | – | – | – | 15 | 1.89 (1.13 to 3.14) |
| Colorectal | 138 | 0.95 (0.80 to 1.12) | 81 | 1.01 (0.81 to 1.26) | 57 | 0.87 (0.67 to 1.13) |
| Breast | 134 | – | – | – | 134 | 1.08 (0.91 to 1.29) |
| Prostate | 79 | – | 79 | 1.21 (0.97 to 1.51) | – | – |
| Bladder | 27 | 0.75 (0.51 to 1.10) | 16 | 0.64 (0.39 to 1.05) | 11 | 1.03 (0.55 to 1.91) |
| Lung and lower airway | 336 | 0.95 (0.85 to 1.05) | 201 | 0.90 (0.78 to 1.03) | 135 | 1.00 (0.84 to 1.19) |
| Oesophagus | 25 | 0.46 (0.31 to 0.68) | 23 | 0.55 (0.37 to 0.83) | <10 | 0.15 (0.04 to 0.62) |
| Corpus uteri | 22 | – | – | – | 22 | 1.72 (1.12 to 2.64) |
| Ovaries | 40 | – | – | – | 40 | 1.19 (0.87 to 1.63) |
| Pancreas | 62 | 0.86 (0.67 to 1.11) | 24 | 0.63 (0.42 to 0.95) | 38 | 0.13 (0.82 to 1.55) |
| Kidney | 20 | 0.61 (0.39 to 0.96) | 14 | 0.70 (0.41 to 1.18) | <10 | 0.46 (0.19 to 1.11) |
| Leukaemia | 48 | 1.21 (0.89 to 1.65) | 25 | 1.17 (0.77 to 1.76) | 23 | 1.27 (0.80 to 2.02) |
| Non-Hodgkin's lymphoma | 48 | 1.29 (0.96 to 1.73) | 35 | 1.60 (1.13 to 2.27) | 13 | 0.84 (0.48 to 1.49) |
| Nervous system | 19 | 0.32 (0.18 to 0.58) | 11 | 0.30 (0.13 to 0.66) | <10 | 0.36 (0.15 to 0.86) |
| All cancers (ICD-10 codes C00–D48) | | 0.98 (0.93 to 1.03) | | 0.92 (0.85 to 0.99) | | 1.05 (0.97 to 1.13) |

N<10 is not shown in accordance with Statistics Netherlands (CBS) guidelines.
ICD-10, 10th revision of the International Statistical Classification of Diseases and Related Health Problems; RR, relative risk.

## Strengths and limitations

This is the first study that investigates cancer mortality specifically among the Moluccan–Dutch, by using historic records on the migration of the Moluccan migrants to the Netherlands, enabling us to select participant based on family name. One limitation of this study is the fact that, for most cancer types, the total number of deaths was small among the Moluccan–Dutch, limiting our ability to detect significant differences between Moluccan–Dutch and the general Dutch population. For the same reason, we had to exclude a separate analysis of third-generation Moluccans.

Another limitation is that we were unable to include second-generation Moluccan–Dutch persons with a Dutch father and consequently Dutch family name. It is estimated that approximately 75% of second-generation Moluccan–Dutch have mixed parents.[12] It is uncertain to what extent our results are representative of all mixed-origin residents. Given that the influence of Moluccan culture often persists in families with one non-Moluccan parent, especially since approximately 40% of mixed-origin residents still live in segregated Moluccan districts, our results may apply, although in attenuated form, to these persons.[12]

## Discussion of the key findings

As cancer risk depends on both factors associated with a migrant's country of origin (eg, high infection pressure)

and also on factors associated with the host country (eg, Western lifestyle), the position of migrant within the host country is an important consideration when interpreting the results. Moluccan–Dutch generally have lower ranking occupations and are lower educated compared with native Dutch.[13] Since we were able to correct for area-level SES in our regression analyses, and this had limited impact on our results, we do not expect socioeconomic differences to have major impact on our results. Another important aspect is migrants' access to healthcare, as detection of cancer in later stages may increase mortality.[14] The evidence on this is equivocal, as Moluccan–Dutch are thought to show lower utilisation of services by general practitioners, yet have similar number of visits to medical specialists, as compared with native Dutch.[15 16]

In the Netherlands, there is one earlier study reporting on cancer mortality among first-generation and second-generation Indonesian immigrants, which includes persons from the Moluccan Islands.[17] There seem to be multiple similarities in cancer mortality disparities between Moluccan–Dutch and the Indonesian immigrant group as a whole in the Netherlands, when compared with their Dutch counterparts. Both groups showed lower stomach cancer mortality, similar breast cancer mortality and higher liver and uterine cancer mortality when compared with their Dutch counterparts. However, the Indonesian immigrant group as a whole

**Table 4** Poisson regression for cancer mortality among Moluccan–Dutch stratified by generation, adjusted for age-at-baseline, sex and socioeconomic status

| | | Generation | | | | |
|---|---|---|---|---|---|---|
| | | 1 (adult) | | 1 (minor) | | 2 |
| Cancer type | N | RR (95% CI) | N | RR (95% CI) | N | RR (95% CI) |
| Stomach | 10 | 0.48 (0.26 to 0.89) | 10 | 0.65 (0.35 to 1.21) | 13 | 1.27 (0.73 to 2.19) |
| Liver | 29 | 3.96 (2.74 to 5.70) | 14 | 1.79 (1.06 to 3.03) | <10 | 1.86 (0.96 to 3.59) |
| Cervix | <10 | 1.72 (0.55 to 5.34) | <10 | 0.91 (0.23 to 3.64) | 10 | 2.39 (1.28 to 4.47) |
| Colorectal | 67 | 1.06 (0.83 to 1.34) | 48 | 0.92 (0.69 to 1.22) | 23 | 0.75 (0.50 to 1.13) |
| Breast | 24 | 0.79 (0.53 to 1.17) | 42 | 1.05 (0.77 to 1.42) | 65 | 1.29 (1.01 to 1.65) |
| Prostate | 71 | 1.61 (1.28 to 2.03) | <10 | 0.45 (0.22 to 0.90) | – | – |
| Bladder | 11 | 0.58 (0.32 to 1.05) | <10 | 0.48 (0.20 to 1.16) | 10 | 1.81 (0.97 to 3.38) |
| Lung and lower airway | 99 | 0.82 (0.68 to 1.01) | 99 | 0.70 (0.58 to 0.85) | 135 | 1.42 (1.20 to 1.69) |
| Oesophagus | <10 | 0.37 (0.17 to 0.82) | <10 | 0.39 (0.21 to 0.76) | 10 | 0.63 (0.34 to 1.16) |
| Corpus uteri | <10 | 1.21 (0.54 to 2.69) | <10 | 1.52 (0.71 to 3.17) | <10 | 3.14 (1.56 to 6.33) |
| Ovaries | <10 | 0.67 (0.32 to 1.40) | 15 | 1.18 (0.71 to 1.96) | 17 | 1.76 (1.09 to 2.83) |
| Pancreas | 29 | 1.09 (0.76 to 1.57) | 23 | 0.81 (0.54 to 1.21) | 10 | 0.59 (0.31 to 1.09) |
| Kidney | <10 | 0.52 (0.23 to 1.15) | <10 | 0.43 (0.18 to 1.04) | <10 | 1.01 (0.50 to 2.02) |
| Leukaemia | 17 | 1.14 (0.71 to 1.84) | <10 | 0.79 (0.41 to 1.51) | 15 | 1.95 (1.17 to 3.25) |
| Non-Hodgkin's lymphoma | 24 | 1.68 (1.13 to 2.51) | 12 | 0.98 (0.55 to 1.72) | <10 | 1.03 (0.51 to 2.07) |
| Nervous system | <10 | 0.36 (0.09 to 1.44) | <10 | 0.39 (0.16 to 0.93) | <10 | 0.25 (0.09 to 0.66) |

N<10 is not shown in accordance with Statistics Netherlands (CBS) guidelines.
RR, relative risk.

showed lower lung and colorectal cancer mortality when compared with native Dutch, whereas we found no differences between Moluccan–Dutch and the general Dutch population for these cancer types.

In Europe, a marked feature of cancer epidemiology in LMIC migrant groups is their lower all-cancer mortality risk.[1] However, we found that the Moluccan–Dutch experience similar all-cancer mortality when compared with the general Dutch population. Even though men showed significantly lower all-cancer mortality when compared with men in the general Dutch population, the difference was small as compared with other LMIC migrants in the Netherlands.[4 6]

Another characteristic of cancer epidemiology in LMIC migrants in Europe is a higher prevalence of cancer related to infection (pathogen-driven) and a lower prevalence of cancers thought to be related to Western lifestyle (non-pathogen driven).[1] Our results imply that the Moluccan–Dutch do not fully adhere to this pattern. We will discuss all cancers for which we found a significant difference between Moluccan–Dutch and the general Dutch population and the cancer types that carried the highest mortality rates among both groups.

For cancers in which exposure to infectious agents is an important risk factor, we found higher mortality risk among Moluccan–Dutch for liver and cervical cancer and a lower mortality risk for stomach cancer, when compared with the general Dutch population.

The higher liver cancer mortality among the Moluccan–Dutch, especially in the first generation, might be reflective of early life exposure in Indonesia to higher burden of hepatitis B and C infection, the most important risk factor for liver cancer.[18] The fact that first-generation migrants from Indonesia in the Netherlands show higher infectious hepatitis mortality than the general Dutch population further supports to this theory.[17] Another important risk factor for liver cancer is alcohol consumption. However, since a health survey among Moluccan–Dutch found their level of alcohol consumption to be lower when compared with native Dutch, this risk factor is unlikely to explain the difference found.[19]

Cervical cancer mortality was only significantly elevated among Moluccan–Dutch women of the second generation, when compared with their counterparts in the general Dutch population. This is in contrast with earlier research on LMIC-migrants in Europe where women of the first generation were at increased risk.[1 5] There is no data on the prevalence among Moluccan–Dutch women of the most important etiological factors for cervical cancer, specific strains of the sexually transmitted human papillomavirus.[20]

The lower stomach cancer mortality among Moluccan–Dutch was mostly attributable to a significantly lower risk in first-generation adults when compared with the general Dutch population. Lower prevalence in Indonesia of the main risk factor, colonisation with the bacteria

Helicobacter pylori, might have had a protective effect in this group.[21 22]

Furthermore, we found that the Moluccan–Dutch carry higher mortality risk for cancer of the corpus uteri, but lower mortality risk for oesophageal and kidney cancer and cancers of the nervous system.

For cancer of the corpus uteri, the higher mortality risk among Moluccan–Dutch women when compared with women from the general Dutch population is probably not explained by an important risk factor, obesity, since obesity rate was found to be similar between the two groups.[19 23] Another important risk factor is (cumulative) oestrogen exposure, but data on this topic in Moluccan–Dutch women are lacking.[24]

For oesophageal cancer, important risk factors are mostly related to Western lifestyle, such as smoking, alcohol consumption and obesity.[25] The Moluccan–Dutch show no significant difference in smoking or body mass index when compared with native Dutch, but they are thought to consume less alcohol, which could contribute to the results.[19]

Kidney cancer mortality risk was lower among Moluccan–Dutch, when compared with the general Dutch population. A known risk factor for kidney cancer is hypertension, which actually may be more prevalent among Moluccan–Dutch.[15 19] Other risk factors are mostly related to a Western lifestyle, such as smoking and obesity, but prevalence of these particular risk factors is thought to be similar between Moluccan–Dutch and native Dutch.[19 26]

Cancer of the nervous system mortality risk was lower among Moluccan–Dutch than among the general Dutch population. To date, only a few factors are known to influence the incidence of gliomas, such as ionising radiation exposure, history of atopic constitution and some genetic risk factors.[27 28] Whether these factors are able to explain the lower risk among the Moluccan–Dutch is unclear.

We found no difference in risk of mortality between the entire group of Moluccan–Dutch and the general Dutch population for the cancer types that carried the highest mortality rates.

In lung cancer, Moluccan–Dutch of the second generation showed a higher mortality risk, when compared with the general Dutch population. Smoking is the most important risk factor of lung cancer.[29] As stated earlier, a health survey among Moluccan–Dutch found no difference in smoking between Moluccan–Dutch and native Dutch, but no stratification by generation was made.[19]

In breast cancer, Moluccan–Dutch women of the second generation experienced increased mortality risk, when compared with the general Dutch population. Since migrant women are thought to show lower attendance at the national breast cancer screening programme, lower detection rate in the early stages of breast cancer, lowering the chance of curable treatment options and consequently increase mortality, might contribute to the results.[14 30] However, this raises the question why first-generation Moluccan–Dutch women do not show higher mortality risk for breast cancer.

For colorectal cancer, there was no difference in the mortality risk between Moluccan–Dutch and the general Dutch population, which is in contrast with most other LMIC migrant groups in Europe.[1] One could speculate that the similar mortality risk detected in our study might be reflective of adaptation to Western lifestyle among the Moluccan–Dutch since 1951.[31]

In conclusion, several decades after migration from Indonesia to the Netherlands, the Moluccan–Dutch showed similar all-cancer mortality when compared with the general Dutch population, but differences exist in cancer-specific mortality between the two groups, in both the first and second generation. In liver and stomach cancer, an increased period of exposure to local risk factors in the Netherlands might have led to a shift in mortality towards rates of the general Dutch population. However, for most other cancer types, such a tendency was not detected, unlike some other studies in the Netherlands that included second-generation migrants.[4 6] These results highlight the need for research aimed at characterising the cancer profile of LMIC migrants in HICs in order to aid tailored preventative and diagnostic efforts.

**Acknowledgements** We thank the Dutch organisation BUAT (Platform for Moluccans) for contributing financially to the acquisition of data from the Dutch national cause of death registry. Furthermore, we thank Statistics Netherlands for their aid in obtaining the mortality and demographic data.

**Contributors** JMvdW contributed to the design of the study, analysed the data presented in this article and wrote the first draft of the article. AB contributed to the design of the study, lead the acquisition of the data, assisted in the analysis of the data and was involved in drafting the manuscript. CA contributed substantially to the drafting of the manuscript. AK contributed to the design of the study, assisted in the analysis of the data and commented on previous drafts of the manuscript.

**Funding statement** This work was supported by the Dutch organisation BUAT (platform for Moluccans) in order to obtain data from the Dutch national cause of death registry, Grant Number CJ2081008.

**Competing interests** None declared.

**Patient consent for publication** Not required.

**Provenance and peer review** Not commissioned; externally peer reviewed.

**Data availability statement** No additional data are available.

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
