## [Reviewer comments · BMJ Open]

ARTICLE DETAILS

TITLE (PROVISIONAL)	A population-based retrospective study comparing cancer mortality between Moluccan migrants and the general Dutch population: equal risk 65 years after immigration?
AUTHORS	van der Wal, Junus; Bodewes, Adeë; Agyemang, Charles; Kunst, Anton

VERSION 1 – REVIEW

REVIEWER	Eva-Maria Berens Bielefeld University, Germany
REVIEW RETURNED	05-Mar-2019

GENERAL COMMENTS	The article aims to investigate cancer mortality among Moluccan-Dutch migrants compared to the general Dutch population. The authors analyse all-cancer mortality and cancer-specific mortality among first and second generation immigrants. Overall this topic is very interesting and more evidence is needed. However, I have some suggestions that might help to further improve the analyses presented. Please find my detailed suggestions below. Introduction I.100: Please avoid formulations such as “ineed”, furthermore there is a typo “compare” needs to be “compared”. I.107: Please give a reference for the first sentence in the paragraph. Please add more details on previous results on cancer mortality among (first and second generation) immigrants (to the Netherlands) also with focus on possible differences in type of cancer and possible convergence of risks over time. Please give more details on the group of Moluccan-Dutch persons in the Netherlands and integrate the first paragraph of the methods section into the Introduction. Methods Please integrate the paragraph on the study population into the introduction as this does not contain information on the methodology, i.e. how this group was operationalized in the data. Information on integration is not needed here but may be helpful in the discussion. Please describe the data sources in more detail and restructure the paragraph: 1. Data source for identification of Moluccans: Who holds the data of the family names registered among Moluccans upon arrival? How many persons were registered in that list? Please also include more information on discrimination of family names of Moluccans compared to other Indonesian migrants and Dutch.
--

	2. Data basis for cancer mortality analysis. Describe the data basis in detail. Which information is included (which not). 3. How was linkage done? Identification of Moluccans by family name. All cases with family names equal to register-source are defined as Moluccans. Furthermore you describe how generation was operationalized. In the results you refer to two types of first generation migrants, which is not included in the description. In the description you refer to third generation, but this is not mentioned in the results. Please reconsider the stratification and description of generation. Given the small n in some cancer entities and the limitation on second generation, I suggest to avoid building a group of third generation. The stratification of minor and adult first generation seems interesting, however the underlying hypothesis is lacking and should be included in the text. But again, as n is small, therefore there should be a strong hypothesis suggesting the stratification. Furthermore the cut-off minors/adults is not explained. Do minors have a different risk of exposure compared to adults unless a longer time of exposure? They also have lived in the country of origin (including habits, hygiene and healthcare etc.) for years. Another thought on generation, I would like you to address is that some factors directly change after migration: e.g. decreased risk for some infections, different hygiene standards. But other aspects such a (food) habits remain after migration and maybe there is adaption of some habits of the general Dutch population. Table 2 needs to be table 1 and vice-versa. Results Please structure the text according to your underlying hypothesis (the difference among infection agent related cancers and lifestyle cancers) in the tables and in the text. Discussion I.262: Please give more details on the limitation to identify second generation immigrants. How high the rate of mixed-marriages among Moluccan immigrants (with Dutch or other Indonesian migrants). Please add information on accuracy of the Dutch cancer registry. Might there be bias in registration of cancer causes? Please also discuss the possible influence of access to health care and prevention on cancer mortality among Moluccans. There is evidence that some migrants are diagnosed with advanced cancer-stages. This might affect the results as higher mortality might not only be a result of differences in infection but also of differences in access to health care.
--	--

REVIEWER	Allison Drosdowsky Peter MacCallum Cancer Centre
REVIEW RETURNED	26-Apr-2019

GENERAL COMMENTS	Thank you for the opportunity to review this manuscript. This is a very interesting piece of research, on a unique population that lends themselves to this kind of study. The introduction and discussion in particular are very thorough, and well referenced in regards to models of cancer risk among migrant groups. I do however have some issues with the description and conduct of the statistical analyses that are detailed below. I think with a more thorough description of how the analyses were performed, these issues could be remedied.
--

	The 'Data sources and handling' section requires more detail in order to understand the data that has been used in these analyses.  - Missing from the 'Data sources and handling' are key details including the time frame used in the sourcing of data, and how age was determined and subsequently used in the models. The detail given to describing socioeconomic status and generation is good, but is needed for all the variables used in analysis. The 'Data analysis' section is missing information required to fully explain the analyses that were performed. The models could be described more thoroughly, specifically the populations, comparison groups and the covariates included. For example:  - The general Dutch population sample described in Table 1 consists of 15 866 538 people whereas the line 180 of the manuscript notes 9 064 169 Dutch people were used in the reference group. If this corresponds to only those born prior to 1970, this needs to be stated but also justified. Are Moluccan-Dutch people born after 1970 included? The sample described in Table 1 needs to correspond to the sample used in analysis. A participant flowchart may be of use here. - Table 3 presents the results of an analysis adjusted for sex (as stated in the title), but also presented separately by sex in columns two and three. Similarly, Table 4 is adjusted for by age (as stated in the title) but each generation is presented separately, and generation could be considered a proxy for age. Details of these models should be included in the 'Data analysis' section of the manuscript for clarity. - When modelling risk of mortality in the Moluccan-Dutch cohort by generation, it is unclear if the comparator is the whole of the general Dutch population or the comparable birth year cohort as specified in Table 1. If it is the latter, is this a suitable comparator, considering migrant generation and year of birth are not representing the same thing. Censoring of numbers smaller than 10 is odd in Tables 3 and 4, since they are all directly calculable from data given. I understand this is usually done for privacy reasons, but that does not seem applicable here. These estimates also have associated relative risks and confidence intervals, so some justification for their inclusion despite the small n should be included. Only results where the confidence interval for the relative risk does not include the null are discussed (with the exception of stomach cancer which is discussed despite the confidence interval for the risk ratio including 1) however these are often very small differences – perhaps a measure of effect size (see attached paper) could be more informative? Formatting issues: there is inconsistent use of a decimal separator (./.) in the tables. Table 1 contains a question mark in the missing value for Dutch general population (31734?) Language issues: In the abstract, the Moluccan-Dutch are stated as being the oldest non-Western migrant group in the Netherlands,
--	--

	while the introduction clarifies that they are the oldest non-Western migrant group since the Second World War.
--	---

VERSION 1 – AUTHOR RESPONSE

Reviewer: 1

Reviewer Name: Eva-Maria Berens

Institution and Country: Bielefeld University, Germany

Please state any competing interests or state 'None declared': none declared

Please leave your comments for the authors below

The article aims to investigate cancer mortality among Moluccan-Dutch migrants compared to the general Dutch population. The authors analyse all-cancer mortality and cancer-specific mortality among first and second generation immigrants. Overall this topic is very interesting and more evidence is needed.

However, I have some suggestions that might help to further improve the analyses presented. Please find my detailed suggestions below.

Introduction

I.100: Please avoid formulations such as “inced”, furthermore there is a typo “compare” needs to be “compared”.

Authors response:

We have rephrased sentences with the word “indeed” and adjusted the typo. Thanks!

I.107: Please give a reference for the first sentence in the paragraph.

Authors response:

We have added a reference to substantiate the content of the paragraph. (Gushulak, MacPherson. The basic principles of migration and health: Population mobility and gaps in disease prevalence. Emerging Themes in Epidemiology 2006)

Please add more details on previous results on cancer mortality among (first and second generation) immigrants (to the Netherlands) also with focus on possible differences in type of cancer and possible convergence of risks over time.

Authors response:

We have shortly described the main findings of the studies we reference in this paragraph, with special focus on convergence of risk over time.

Please give more details on the group of Moluccan-Dutch persons in the Netherlands and integrate the first paragraph of the methods section into the Introduction.

Authors response:

In response to this request, we have deleted the section ‘study population’ from the method section, and instead integrated information on the (migration) history of the Moluccan-Dutch in the Introduction and information on adaptation to Dutch society (including healthcare use) in the discussion.

Methods

Please integrate the paragraph on the study population into the introduction as this does not contain information on the methodology, i.e. how this group was operationalized in the data. Information on integration is not needed here but may be helpful in the discussion.

Authors response:

In response to this request, we have deleted the section 'study population' from the method section, and instead integrated information on the (migration) history of the Moluccan-Dutch in the Introduction and information on adaptation to Dutch society (including healthcare use) in the discussion.

Please describe the data sources in more detail and restructure the paragraph:

1. Data source for identification of Moluccans: Who holds the data of the family names registered among Moluccans upon arrival? How many persons were registered in that list? Please also include more information on discrimination of family names of Moluccans compared to other Indonesian migrants and Dutch.

Authors response:

In the revised manuscript, we have explained the creation of the list of family names and identification of Moluccans in registries more in detail.

2. Data basis for cancer mortality analysis. Describe the data basis in detail. Which information is included (which not).

Authors response:

We have added more explanation on the Dutch cause of death registry and what type of data this registry holds.

3. How was linkage done? Identification of Moluccans by family name. All cases with family names equal to register-source are defined as Moluccans.

Authors response:

At Statistics Netherlands, the linkage between death records and population registry is made on the basis of personal identification numbers that are unique to each registered person, this is now clarified in the methods section.

Furthermore you describe how generation was operationalized. In the results you refer to two types of first generation migrants, which is not included in the description. In the description you refer to third generation, but this is not mentioned in the results. Please reconsider the stratification and description of generation. Given the small n in some cancer entities and the limitation on second generation, I suggest to avoid building a group of third generation. The stratification of minor and adult first generation seems interesting, however the underlying hypothesis is lacking and should be included in the text. But again, as n is small, therefore there should be a strong hypothesis suggesting the stratification. Furthermore the cut-off minors/adults is not explained. Do minors have a different risk of exposure compared to adults unless a longer time of exposure? They also have lived in the country of origin (including habits, hygiene and healthcare etc.) for years. Another thought on generation, I would like you to address is that some factors directly change after migration: e.g. decreased risk for some infections, different hygiene standards. But other aspects such as (food) habits remain after migration and maybe there is adaptation of some habits of the general Dutch population.

Authors response:

We chose to stratify first generation minors and adults, since we believe the first generation minors have experienced exposure to cancer risk factors that is distinctly in-between the first and second

generation. Similar to first generation adults, they were born in Indonesia, but they experienced important developmental stages in their life in the Netherlands, were more likely to attend the Dutch education system, and more likely to engage with native Dutch (either friendships or relationships), all of which is expected to influence the level of acculturation/integration and exposure to (lifestyle-related) cancer risk factors such as diet, alcohol consumption and smoking. Meanwhile, the second generation obviously shows overlap with regard to growing up in the host country, but they lack early life exposures to (infection-related) cancer risk factors.

In the revised manuscript, we have explained why we chose to make this stratification.

Table 2 needs to be table 1 and vice-versa.

Authors response:

In the revised manuscript, table 1 is now the first that is mentioned in the text, since we prefer the first table to lay out the characteristics of the study population.

Results

Please structure the text according to your underlying hypothesis (the difference among infection agent related cancers and lifestyle cancers) in the tables and in the text.

Authors response:

Even though exposure to infectious agents and adaptation to Western lifestyle are key elements of our discussion, the main hypothesis is that cancer risk is similar between Moluccan-Dutch and the general Dutch population. Therefore, we prefer to structure the main results according to differences from the main hypothesis (i.e. higher or lower mortality), while separately discussing cancers related to infection in the discussion. Furthermore, to highlight this choice, we put less emphasis on the distinction between pathogen vs. non-pathogen driven cancers in the discussion, but instead only mention the infection related cancers first, followed by the other cancers.

Discussion

I.262: Please give more details on the limitation to identify second generation immigrants. How high the rate of mixed-marriages among Moluccan immigrants (with Dutch or other Indonesian migrants). Please add information on accuracy of the Dutch cancer registry. Might there be bias in registration of cancer causes?

Authors response:

In the limitations section, we added more information on mixed-marriages among the Moluccan-Dutch and explained how this might influence the results.

We are not aware of any bias in the registration of cancer as a cause of death in The Netherlands (other than the well-known limitations of routine cause-of-death registries based on death certificates). All registries used in this study contain all legal residents of the Netherlands.

Please also discuss the possible influence of access to health care and prevention on cancer mortality among Moluccans. There is evidence that some migrants are diagnosed with advanced cancer-stages. This might affect the results as higher mortality might not only be a result of differences in infection but also of differences in access to health care.

Authors response:

We agree with the reviewer and have incorporated the topic of access to health care in the paragraph concerning adaptation of Moluccan-Dutch to Dutch society in the Discussion.

Reviewer: 2

Reviewer Name: Allison Drosdowsky
Institution and Country: Peter MacCallum Cancer Centre
Please state any competing interests or state 'None declared': None declared

Please leave your comments for the authors below

Thank you for the opportunity to review this manuscript. This is a very interesting piece of research, on a unique population that lends themselves to this kind of study. The introduction and discussion in particular are very thorough, and well referenced in regards to models of cancer risk among migrant groups.

I do however have some issues with the description and conduct of the statistical analyses that are detailed below. I think with a more thorough description of how the analyses were performed, these issues could be remedied.

The 'Data sources and handling' section requires more detail in order to understand the data that has been used in these analyses.

- Missing from the 'Data sources and handling' are key details including the time frame used in the sourcing of data, and how age was determined and subsequently used in the models. The detail given to describing socioeconomic status and generation is good, but is needed for all the variables used in analysis.

Authors response:

We have added the requested information in the Methods section of the revised manuscript.

The 'Data analysis' section is missing information required to fully explain the analyses that were performed. The models could be described more thoroughly, specifically the populations, comparison groups and the covariates included. For example:

- The general Dutch population sample described in Table 1 consists of 15 866 538 people whereas the line 180 of the manuscript notes 9 064 169 Dutch people were used in the reference group. If this corresponds to only those born prior to 1970, this needs to be stated but also justified. Are Moluccan-Dutch people born after 1970 included? The sample described in Table 1 needs to correspond to the sample used in analysis. A participant flowchart may be of use here.

Authors response:

We have clarified the text of the Method section, explaining more in detail which groups were used in which analyses were performed and why.

- Table 3 presents the results of an analysis adjusted for sex (as stated in the title), but also presented separately by sex in columns two and three. Similarly, Table 4 is adjusted for by age (as stated in the title) but each generation is presented separately, and generation could be considered a proxy for age. Details of these models should be included in the 'Data analysis' section of the manuscript for clarity.

Authors response:

Results in table 3 are indeed not adjusted for sex, we corrected this in the title of the table. Results in table 4 are adjusted for age-at-baseline (as continuous variable), as age can differ substantially within generations.

- When modelling risk of mortality in the Moluccan-Dutch cohort by generation, it is unclear if the comparator is the whole of the general Dutch population or the comparable birth year cohort as

specified in Table 1. If it is the latter, is this a suitable comparator, considering migrant generation and year of birth are not representing the same thing.

Authors response:

We have clarified the text of the Method section, explaining more in detail which groups were used in which analyses were performed and why. As both the first and second generation Moluccan-Dutch cover specific age groups (given that these generations are defined in terms of year of birth), these generations could only be compared to persons from the general Dutch population in the same age groups.

Censoring of numbers smaller than 10 is odd in Tables 3 and 4, since they are all directly calculable from data given. I understand this is usually done for privacy reasons, but that does not seem applicable here. These estimates also have associated relative risks and confidence intervals, so some justification for their inclusion despite the small n should be included.

Authors response:

Guidelines by Statistics Netherlands (CBS) state that we can't report absolute numbers of events smaller than 10. In the revised manuscript, we have explained this at the bottom of the table, in addition to the Methods section.

Only results where the confidence interval for the relative risk does not include the null are discussed (with the exception of stomach cancer which is discussed despite the confidence interval for the risk ratio including 1) however these are often very small differences – perhaps a measure of effect size (see attached paper) could be more informative?

Authors response:

We think that the presented relative risks are adequate measures of effect size. All differences we discuss are >10%, which we consider important differences at the level of national (sub-)populations. However, we also present smaller differences for breast, colorectal and lung cancer. These cancers are discussed because they carry the highest mortality burden in the Moluccan-Dutch population.

Formatting issues: there is inconsistent use of a decimal separator (./.) in the tables. Table 1 contains a question mark in the missing value for Dutch general population (31734?)

Authors response:

We have corrected these formatting issues, thanks!.

Language issues: In the abstract, the Moluccan-Dutch are stated as being the oldest non-Western migrant group in the Netherlands, while the introduction clarifies that they are the oldest non-Western migrant group since the Second World War.

Authors response:

We have corrected these language issues.

VERSION 2 – REVIEW

REVIEWER	Allison Drosdowsky Peter MacCallum Cancer Centre, Australia
REVIEW RETURNED	26-Jun-2019

GENERAL COMMENTS	Thank you for your response to my queries. The added information in the methodology section has clarified the study processes, and explained the statistical processes. I do have a few minor concerns from the amendments: -In the additional information provided in the methodology section, the authors note that mortality data from 2000-2013 was used in the study (line 152). This has introduced a significant limitation to the results that needs to be discussed. Given the population consisted, in part, of adults who migrated in 1951, only the younger migrants who then survived into late adulthood are able to be included in your analysis. (For example, a hypothetical Moluccan-Dutch person who immigrated aged 35, and died of cancer before the age of 84 would not be included in this study)- In the description of the analysis of mortality by generation (Table 4), the authors have added that age-at-baseline was included as a covariate (Table 4 and also line 196). What does 'baseline' refer to in this context?
---